# Relative Localization within a Quadcopter Unmanned Aerial Vehicle Swarm Based on Airborne Monocular Vision

Xiaokun Si [1], Guozhen Xu [1], Mingxing Ke [1], Haiyan Zhang [1], Kaixiang Tong [2] and Feng Qi [1,*]

1 College of Electronic Engineering, National University of Defense Technology, Hefei 230031, China
2 Beijing Space Information Relay and Transmission Technology Center, Beijing 102300, China
* Correspondence: qifeng17@nudt.edu.cn

**Abstract:** Swarming is one of the important trends in the development of small multi-rotor UAVs. The stable operation of UAV swarms and air-to-ground cooperative operations depend on precise relative position information within the swarm. Existing relative localization solutions mainly rely on passively received external information or expensive and complex sensors, which are not applicable to the application scenarios of small-rotor UAV swarms. Therefore, we develop a relative localization solution based on airborne monocular sensing data to directly realize real-time relative localization among UAVs. First, we apply the lightweight YOLOv8-pose target detection algorithm to realize the real-time detection of quadcopter UAVs and their rotor motors. Then, to improve the computational efficiency, we make full use of the geometric properties of UAVs to derive a more adaptable algorithm for solving the P3P problem. In order to solve the multi-solution problem when less than four motors are detected, we analytically propose a positive solution determination scheme based on reasonable attitude information. We also introduce the maximum weight of the motor-detection confidence into the calculation of relative localization position to further improve the accuracy. Finally, we conducted simulations and practical experiments on an experimental UAV. The experimental results verify the feasibility of the proposed scheme, in which the performance of the core algorithm is significantly improved over the classical algorithm. Our research provides viable solutions to free UAV swarms from external information dependence, apply them to complex environments, improve autonomous collaboration, and reduce costs.

**Keywords:** UAV swarm; relative localization; Perspective-n-Point; GNSS-denied environments; YOLO; keypoint detection

## 1. Introduction

Small multi-rotor UAVs have the advantages of good maneuverability, rich expansion functions, and great intelligence potential, but the limited performance of a single aircraft and poor survivability have also been exposed in use [1]. Swarming can compensate for the weaknesses of a single UAV while further leveraging its strengths [2]. Currently, UAV swarms have shown great value and potential in missions such as aerial Internet of Things (IoT) [3,4], relay communication support [5,6], aerial light shows, regional security [7], and military operations [8], which have become one of the inevitable trends in the development of UAV applications. Accurate real-time position information is the basis for UAVs to accomplish a variety of air-to-ground missions. In addition to absolute position information, it also involves the relative position relationship between each UAV within a swarm. It is no exaggeration to say that relative location information is no less important than absolute location information from a swarm perspective. It enables UAVs to maintain planned formations, avoid collisions with each other, and accomplish coordinated maneuvers [9]. Therefore, precise relative localization is a must for swarm UAVs, which is of great significance in reducing the swarm's reliance on absolute position information and improving the swarm's ability to survive in hazardous environments.

In recent years, solutions based on various hardware and methods have emerged for relative localization problems. While they show good performance, the different characteristics and conditions of use make many of these solutions inappropriate for small multi-rotor UAV swarms. Currently, the acquisition of relative localization information between UAVs still relies heavily on the absolute position data of each UAV from the Global Navigation Satellite System (GNSS) [10]. In addition, similar problems exist with relative localization via motion capture systems, simultaneous localization and mapping (SLAM) [11,12], and ground-based ultra-wide band (UWB) localization systems [13]. They all need to first obtain their respective position coordinates in the same spatial coordinate system from external infrastructure or environmental information and then solve for the relative localization information based on this. These methods have obvious drawbacks. Firstly, once absolute localization has failed, relative localization will also not be possible, for example, when encountering a GNSS-denied environment, when the coverage of ground-based localization stations is exceeded or when the environmental features required for SLAM are not evident. Secondly, errors in absolute localization will be superimposed and magnified during the conversion to relative localization information [14]. In addition, absolute localization will take up limited resources per swarm UAV, which could have been avoided.

The model for UAV swarms is derived from the group behavior of flying creatures in nature [15]. They usually rely on organ functions such as vision and hearing to directly obtain information about their relative positions to each other. UAV swarms, as multi-intelligence systems, should also have the ability to achieve relative localization without relying on external facilities or information. Similar functions have already been implemented in the rapidly developing field of advanced driving assistance system (ADAS) research [16,17]. Based on the information provided by vision, laser, and other sensors, it has been possible to achieve accurate relative positioning of objects within a certain range while the vehicle is in motion. However, the environment in which vehicles are driven can be approximated as a two-dimensional space, whereas drones are in a more complex three-dimensional scenario.

Relative localization based on radio signals is a classical approach, currently represented by airborne UWB and relative localization based on carrier phase [18,19]. Although they are superior in terms of localization accuracy, they will significantly increase the cost, power consumption, and system complexity of each UAV, as well as taking into account mutual interference problems. While LIDAR has superior performance and proven applications, the same expensive price and high power consumption prevent it from being the first choice for swarm UAVs [20]. Millimeter-wave radar is less expensive, but it has lower localization accuracy and a smaller measurement range [21].

While relative localization achieved based on vision SLAM is not considered due to its indirectness and instability, vision sensors can also directly provide useful information for relative localization [22]. Wide-angle lenses, gimbals, camera scheduling algorithms, and target tracking algorithms [23] ensure flexible acquisition of environmental images [24]. Binocular cameras and depth cameras are the current mainstream vision solutions [25]. Binocular vision localization uses the principle of triangular geometric parallax to achieve relative localization. However, the co-processing of binocular data requires high computing resources and speed, and the accuracy and range of measurements are limited when the parallax is small. Depth cameras can obtain depth data based on the principle of structured light or time of flight (ToF), but they have a relatively small applicable distance and imaging field of view, making them unsuitable for the relative localization of drones in motion [26].

Monocular cameras are common onboard sensors for UAVs and have the advantage of being cheap and easy to deploy. However, information based solely on a single frame from a single camera can only measure direction but not distance unless more auxiliary information is introduced, which is also the core problem that needs to be solved for monocular visual localization [27]. The implementation of relative localization based on airborne monocular vision offers significant advantages in terms of cost, complexity, and

hardware requirements compared to the other methods mentioned above, but there is a lack of mature solutions. Therefore, the development of a relative localization method based only on airborne monocular vision is of great practical importance to solve the relative localization problem of small multi-rotor UAV swarms.

In this research, we develop an airborne monocular-vision-based relative localization scheme using a small quadrotor UAV as an experimental platform. It achieves accurate real-time relative localization between UAVs based only on a single airborne camera's data and simple feature information of the quadrotor UAV. In summary, our contributions are as follows:

- We propose a new idea of directly using only the rotor motors as the basis for localization and use the deep-learning-based YOLOv8-pose keypoint detection algorithm to achieve fast and accurate detection of UAVs and their motors. Compared to other visual localization information sources, we do not add additional conditions and data acquisition is more direct and precise.
- A more suitable algorithm for solving the PnP (Perspective-n-Point) problem is derived based on the image plane 2D coordinates of rotor motors and the shape feature information of the UAV. Our algorithm is optimized for the application target, reduces the complexity of the algorithm by exploiting the geometric features of the UAV, and is faster and more accurate than classical algorithms.
- For the multi-solution problem of P3P, we propose a new scheme to determine the unique correct solution based on the pose information instead of the traditional reprojection method, which solves the problem of occluded motors during visual relative localization. The proposed method breaks the limitations of classical methods and reduces the amount of data necessary for visual localization.

A description of symbols and mathematical notations involved in this paper is shown in Table 1.

**Table 1.** Description of symbols and mathematical notations.

| | |
|---|---|
| $\{A_i\}$ | The set of points corresponding to all values of $i$. |
| $(a, b)$ | Coordinates in the specified coordinate system. |
| $Oxyz$ | The spatial coordinate system with $O$ as the origin and $Ox$, $Oy$ and $Oz$ as the positive directions of the coordinate axes. |
| $\angle AOB$ | The angle between the rays $OA$ and $OB$ with $O$ as the vertex. |
| $A$ | Matrices, including vectors. |
| $AB$ | A vector with $A$ as the starting point and $B$ as the ending point. |
| $t_n^m$ | The displacement matrix of the $O_m$-coordinate system with respect to the $O_n$-coordinate system. |
| $R_n^m$ | The rotation matrix of the $O_m$-coordinate system with respect to the $O_n$-coordinate system. |
| $A \times B$ | Multiply matrix $A$ with matrix $B$. |
| $[\cdot]^T$ | The transpose of the matrix. |
| $\|\cdot\|$ | The modulus of the vector. |

## 2. Related Work

### 2.1. Monocular Visual Localization

Currently, the main specific methods for monocular visual localization are feature point methods, direct methods, deep-learning-based methods, and semantic-information-based methods. References [28,29] both propose the use of deep learning target detection algorithms to classify and detect images from different angles of the UAV and then combine this with the corresponding dimensional information to estimate the relative position of the UAV. However, this places high demands on the detection model; an accurate detection model often means a larger amount of data collection for training as well as slower detection speeds, while simplifying the model will lead to a significant increase in error. Another

idea is to artificially add features to the UAV to aid detection. In reference [30], Zhao et al. used the derived P4P algorithm to solve the relative position information of the target UAV based on the image positions of four LEDs pre-mounted on the UAV, but only semi-physical simulation experiments were carried out. Walter et al. obtained real-time relative position information of the UAV by detecting scintillating UV markers added to the UAV and using a 3D time-position Hough transform [31]. In reference [32], Saska et al. achieved relative localization in their study by deploying geometric patterns on the UAV and detecting them, with the study also incorporating inertial guidance information. Zhao et al. instead used the April Tag algorithm to achieve the acquisition of UAV position and attitude information by detecting and processing the onboard 2D code [33]. While these methods can achieve good results, the additional addition of features is not conducive to practical application and is not a preferred option. In reference [34], Pan et al. propose a learning-based correspondence point matching model to solve the position information of ground targets based on multiple frames from the UAV's onboard monocular camera. But this method is based less on real time and cannot adapt to the high-speed movement characteristics of UAVs. Reference [35] presents a method for obtaining UAV position and attitude information by inspecting the four rotor motors and other key components of the UAV and applying an improved PnP algorithm. However, we do not believe it is possible to detect so many characteristics of a UAV at the same time when detecting it in the air.

Based on the above analysis, harsh condition constraints, higher acquisition difficulty, and lower real-time and accuracy are the main problems in acquiring data sources for visual localization. We believe that relative localization based on the image feature information of the UAV itself is a feasible idea. Moreover, the number of feature points should be required to be as small as possible to facilitate detection and fast solving. The rotor motors are a necessary component of a quadcopter drone, and there are at least three of them visible when viewed from almost any angle. Therefore, we consider the motors as a reference point for visual localization and explore solving the PnP problem based on better parameters and computational effort.

### 2.2. Target and Keypoint Detection

Accurate detection of the UAV and its motors is the basis for visual localization. Deep-learning-based target detection algorithms are the current mainstream solution, with representative algorithms such as Faster R-CNN, YOLO, and SSD. Compared to other algorithms, the YOLO algorithm is based on the idea of one-off detection, which is faster to process and more suitable for applications in real-time scenarios [36]. Thanks to the simple network architecture and optimized algorithm design, the YOLO algorithm is simple to deploy and more conducive to deployment on lower-performance edge computers. Based on these advantages, the YOLO algorithm is widely used in ground-to-UAV and UAV-to-ground target detection in real time. However, detection accuracy, localization precision, and performance on small targets have been the relative disadvantages of the YOLO algorithm and have been the focus of its iteration and improvement [37].

The YOLO algorithm has now evolved to the latest v8 version, with many improvements referencing the strengths of previous versions. YOLOv8 improves on the FPN (feature pyramid networks) idea and the Darknet53 backbone network by replacing the C3 structure in YOLOv5 with the more gradient flow-rich C2f structure. This improves the multi-scale predictive capability and lightness of the algorithm. In the Head section, YOLOv8 uses the mainstream decoupled head structure and replaces Anchor-Base with Anchor-Free. in addition, YOLOv8 is optimized for multi-scale training, data enhancement, and post-processing optimization, making it easier to deploy and train [38]. The YOLOv8 development team has also released a pre-trained human pose detection model, YOLOv8-pose, as seen in reference [39]. Pose estimation is realized based on the detection and localization of specific parts and joints of the human body. Therefore, YOLOv8-pose can be considered as a method for keypoint detection [40].

Previous related work has focused on detecting UAV motors as area targets based on their additional characteristics [30,31,35]. In this study, we apply YOLOv8-pose, which is used for human posture detection, to the detection of the motors of UAVs. We hope to realize direct, accurate, and real-time access to localization data sources based on the advantages of YOLOv8-pose.

*2.3. Solving the PnP Problem*

The PnP problem is one of the classic problems in computer vision. It involves determining the position and orientation of a camera, given *n* points in three-dimensional space and their corresponding projection points on the camera image plane, combined with the camera parameters. Common solution methods include Gao's P3P [41], direct linear transformation (DLT) [42], EPnP (Efficient PnP) [43], UPnP (uncalibrated PnP) [44], etc. They have different requirements for the number of 2D–3D point pairs and are suitable for different scenarios. In practice, there are often errors in the coordinates of the projected points. More point pairs tend to help improve the accuracy and robustness of the results but increase the amount of work involved in matching and solving the point pairs. Due to the occurrence of occlusion, when photographing another quadcopter UAV with the onboard camera, often only three motors are detected. Three sets of point pairs are also the minimum requirement for solving the PnP problem, also known as the P3P problem.

Current solution methods for P3P problems can be divided into two-stage methods and single-stage methods. The classical Gao's method [41] mainly uses similar triangles, the cosine theorem, and Wu's elimination method to solve the problem. In reference [45], Li et al. proposed a geometric feature based on a perspective similar triangle (PST), reducing the unknown parameters, reducing the complexity of the equations, and showing a more robust performance. However, they all require the distance from the camera to the three points to be found first, and then use methods such as singular value decomposition (SVD) to obtain position and pose information. The single-stage method eliminates the intermediate process of solving for distance values, which is more in line with the application needs of this study. The method proposed by Kneip is representative of the single-stage method, which derives the solution for camera position and pose directly by introducing an intermediate camera and a series of geometrical treatments [46]. It offers a significant speed improvement over Gao's method, although at the cost of complex geometric transformations. Furthermore, all P3P solutions mention the need to deal with the non-uniqueness of the solution of the P3P problem by the reprojection method using the fourth set of point pairs. However, in reality, when viewed from a partial angle, only three motors are often observable due to the fuselage's shading.

Classical PnP solution methods are devoted to solving general problems and do not satisfy the special cases in this study. Meanwhile, more geometric features of rotor UAVs are not utilized in these methods. In this research, we follow the idea of the single-stage method and derive the position result of the P3P problem directly from an algebraic resolution perspective based on the dimensional characteristics of the quadrotor UAV. For the multi-solution problem of P3P, we propose a solution that does not require a fourth set of point pairs based on the attitude characteristics of the UAV.

## 3. Detection of UAVs and Motors

### 3.1. Detection Model Training

First, we simulate the perceptual behavior of on-board vision by photographing a quadrotor UAV hovering in the air from different angles and distances, as shown in Figure 1. We then label the captured images, where UAVs are labeled as detection targets with rectangles and motors are labeled as keypoints with dots. In order to correctly correspond to the 2D–3D point pairs, the motor labeling order is specified as clockwise from the first motor on the left, viewed from the bottom up. Obscured motors are not labeled. Finally, following the general steps of YOLOv8-pose model training, the labeled images and data were imported to generate the training model.

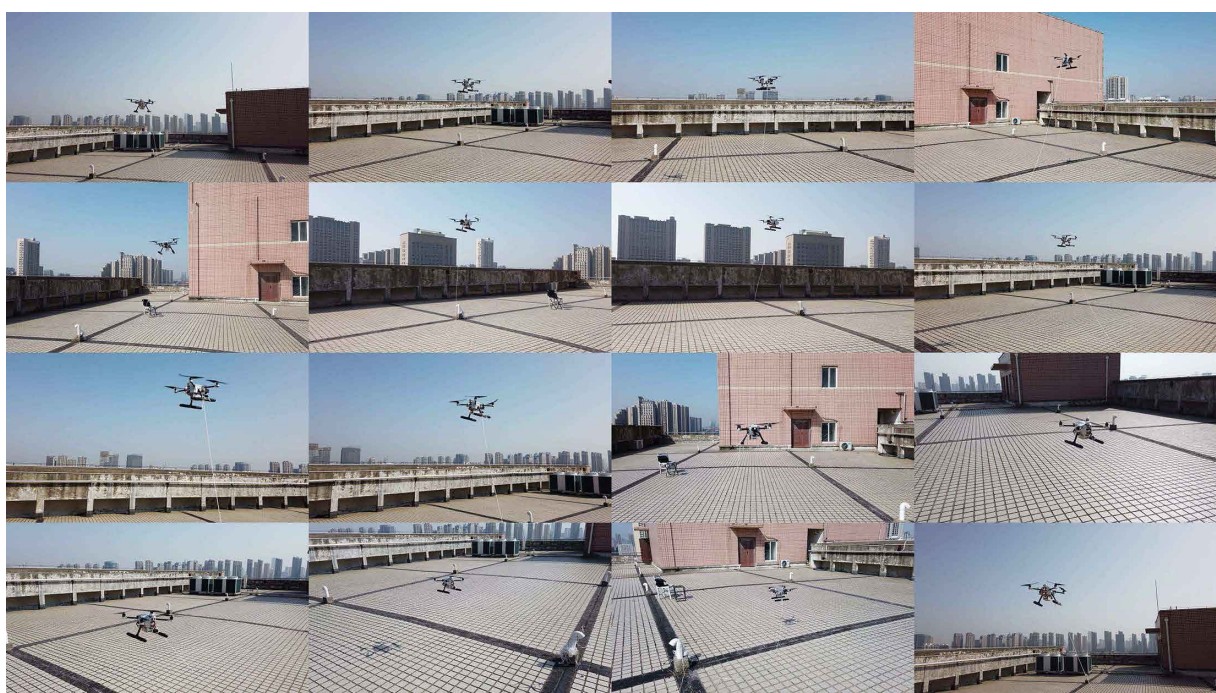

**Figure 1.** Acquisition of UAV images.

### 3.2. Sequencing of Motor Keypoints

Although the labeling order of the motors has been specified, the output order of the motor keypoints may still be wrong due to the complexity of the UAV's flight attitude and the multiple angles of detection. Therefore the sequence of keypoints of motors needs to be calibrated. Due to the presence of occlusion, two to four motors can be detected in one frame, as shown in Figure 2.

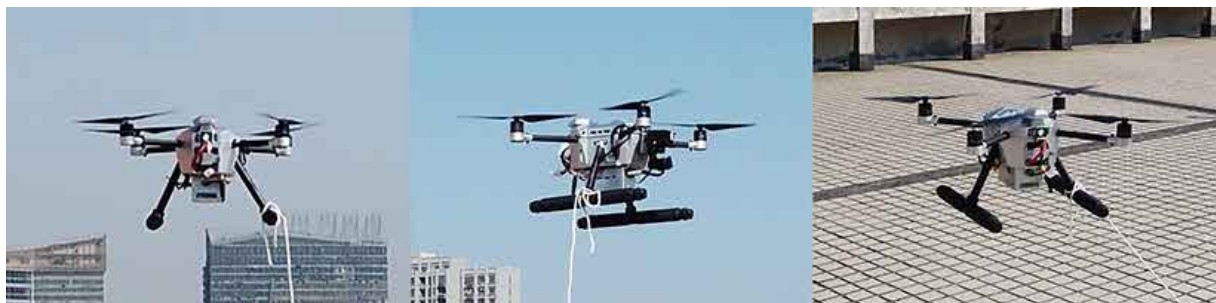

**Figure 2.** Three cases of the number of motors can be seen.

We set the pixel coordinates of the motors on the image plane to be $\{P_i^0 = (u_i^0, v_i^0)\}$ ($i = 1,2,3,4$), and the correct coordinates after sorting to be $\{P_i = (u_i, v_i)\}$. When two to three motors can be detected, we specify that the motors appearing on the screen are sorted from left to right. When all four motors are detected, we use the condition that the two midpoints of the lines connecting the non-adjacent motors should theoretically overlap to judge and correct the motor order. The specific algorithm for sorting is shown in Algorithm 1:

---

**Algorithm 1** Sorting the four motors

---

**Require:** $\{P_i^0 = (u_i^0, v_i^0)\}, i \in \{1:n\}$
**Ensure:** $\{P_i = (u_i, v_i)\}$

1: **if** $n < 4$ **then**
2:      Sort $P_{1:n}^0$ by $u_1^0 < u_2^0 \; (< u_3^0)$
3: **else**
4:      **for** $i, j \in \{1:n\}, i < j$ **do**
5:          $o_{ij} = [\frac{u_i^0 + u_j^0}{2}, \frac{v_i^0 + v_j^0}{2}]$
6:      **end for**
7:      $d_1 = \|o_{12}o_{34}\|, d_2 = \|o_{13}o_{24}\|, d_1 = \|o_{14}o_{23}\|$
8:      **if** $min\{d_{1:3}\} = d_1$ **then**
9:          Swap the values of $P_2^0$ and $P_3^0$
10:      **else if** $min\{d_{1:3}\} = d_3$ **then**
11:          Swap the values of $P_3^0$ and $P_4^0$
12:      **end if**
13: **end if**
14: $\{P_i\} = \{P_i^0\}$

---

## 4. Relative Position Solution Method

### 4.1. Problem Model

Typically, the onboard vision sensor can detect three to four motors of the UAV within the field of view. The solution of the relative position at this point is a P3P problem.

The model of the P3P problem is shown in Figure 3. Camera coordinate system, pixel coordinate system, and motor coordinate system are established separately. $O_c$ is the optical centre of the camera and $O_p uv$ is the pixel coordinate system. The right-angle coordinate system $O_c x_c y_c z_c$ is established with $O_c$ as the origin, where the $x_c$-axis is in the same direction as the $u$-axis, the $z_c$-axis is reversed with the $v$-axis, and the $y_c$-axis is on the optical axis. $\{M_i\}(i = 1, 2, 3, 4)$ represents the four motors of the UAV and $O_m$ is the intersection of the central axis of the UAV with the plane where the motors are located, here representing the spatial position of the UAV. We set up the right-angle coordinate system $O_m x_m y_m z_m$ with the point $O_m$ as the origin, where the $x_m$-axis and $y_m$-axis are in the positive direction of $O_m M_3$ and $O_m M_4$, respectively, and the $z_m$-axis points above the top of the UAV.

In fact, the camera coordinate system and the motor coordinate system express the motion attitude of the camera gimbal and the UAV, which can be understood as the result of the transformation with respect to the Earth coordinate system or the inertial coordinate system. The pixel coordinate system is fixed with respect to the camera coordinate system and is determined by the internal parameters of the camera. Then, the P3P problem is converted to solving for the translation $t_c^m$ and rotation $R_c^m$ of the motors coordinate system with respect to the camera coordinate system, which are set as

$$t_c^m = \begin{bmatrix} t_x \\ t_y \\ t_z \end{bmatrix}, \quad R_c^m = \begin{bmatrix} r_{11} & r_{12} & r_{13} \\ r_{21} & r_{22} & r_{23} \\ r_{31} & r_{32} & r_{33} \end{bmatrix}, \tag{1}$$

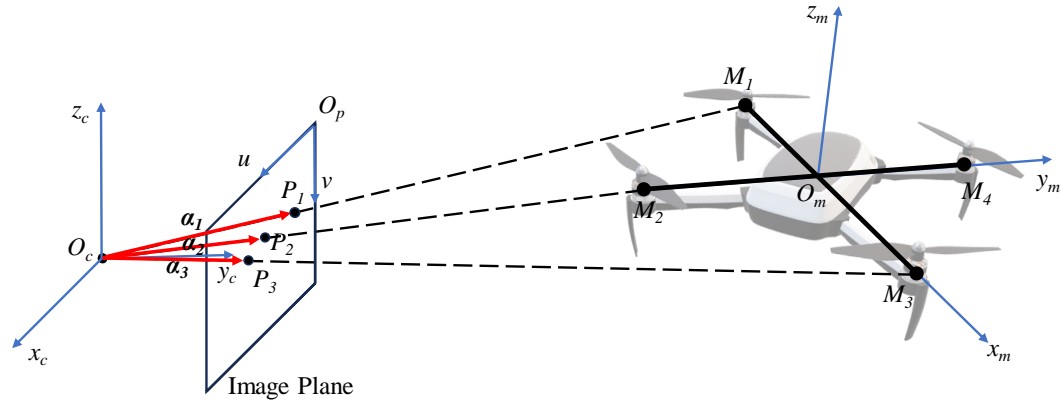

**Figure 3.** The model for the P3P problem.

### 4.2. Improved Solution Scheme for the P3P Problem

We first consider the general case where only three motors are detected. The pixel coordinates $P_i$ of the motors and the camera focal length $f$ are known. The vectors $\boldsymbol{\alpha}_i$ represent $\boldsymbol{O_c P_i}$. Obviously,

$$\boldsymbol{\alpha}_i = [u_i^c, \ f, \ v_i^c]^{\mathrm{T}}, \ i = 1, 2, 3, \tag{2}$$

where

$$\begin{cases} u_i^c = \dfrac{u_i - \frac{W_p}{2}}{\frac{W_p}{2}} \cdot \dfrac{W_I}{2}, \\ v_i^c = -\dfrac{v_i - \frac{H_p}{2}}{\frac{H_p}{2}} \cdot \dfrac{H_I}{2}, \end{cases} \tag{3}$$

where $W_p$ and $H_p$ represent the pixel width and height of the image plane, and $W_I$ and $H_I$ represent the actual width and height of it.

Obviously, the point $P_i$ is the projection on the image plane of the reflected rays from the point $M_i$ when they strike the focal point $O_c$ along a straight line. So, $\boldsymbol{O_c M_i}$ can be expressed as

$$\boldsymbol{O_c M_i} = k_i \boldsymbol{\alpha}_i, \ i = 1, 2, 3. \tag{4}$$

We set $\|\boldsymbol{O_m M_i}\| = d$, which can be obtained by measuring. Accordingly,

$$\begin{aligned} \boldsymbol{O_m M_1} &= [-d, \ 0, \ 0]^{\mathrm{T}}, \\ \boldsymbol{O_m M_2} &= [\ 0, -d, \ 0]^{\mathrm{T}}, \\ \boldsymbol{O_m M_3} &= [\ d, \ 0, \ 0]^{\mathrm{T}}. \end{aligned} \tag{5}$$

Based on the rules of vector transformation, $\boldsymbol{O_c M_i}$ can also be obtained from $\boldsymbol{O_m M_i}$ by the following transformation,

$$\boldsymbol{O_c M_i} = \boldsymbol{R_c^m} \times \boldsymbol{O_m M_i} + \boldsymbol{t_c^m}, \ i = 1, 2, 3. \tag{6}$$

From (1), (4), (5), and (6), it follows that

$$k_1 \boldsymbol{\alpha_1} = - \begin{bmatrix} r_{11} \\ r_{21} \\ r_{31} \end{bmatrix} d + \boldsymbol{t_c^m},$$

$$k_2 \boldsymbol{\alpha_2} = - \begin{bmatrix} r_{12} \\ r_{22} \\ r_{32} \end{bmatrix} d + \boldsymbol{t_c^m}, \tag{7}$$

$$k_3 \boldsymbol{\alpha_3} = \begin{bmatrix} r_{11} \\ r_{21} \\ r_{31} \end{bmatrix} d + \boldsymbol{t_c^m}.$$

To eliminate the unknown quantity $k_i$, the first and second rows of each equation in (7) are divided by the third row, respectively, and substitute (2), thus obtaining

$$\frac{-r_{11}d + t_x}{-r_{31}d + t_z} = \frac{u_1^c}{v_1^c}, \quad \frac{-r_{21}d + t_y}{-r_{31}d + t_z} = \frac{f}{v_1^c},$$

$$\frac{-r_{12}d + t_x}{-r_{32}d + t_z} = \frac{u_2^c}{v_2^c}, \quad \frac{-r_{22}d + t_y}{-r_{32}d + t_z} = \frac{f}{v_2^c}, \tag{8}$$

$$\frac{r_{11}d + t_x}{r_{31}d + t_z} = \frac{u_3^c}{v_3^c}, \quad \frac{r_{21}d + t_y}{r_{31}d + t_z} = \frac{f}{v_3^c}.$$

Then, divide both the numerator and denominator on the left side of the Equation (8) by $t_z$, and we can obtain

$$\frac{-r_{11}d/t_z + t_x/t_z}{-r_{31}d/t_z + 1} = \frac{u_1^c}{v_1^c}, \quad \frac{-r_{21}d/t_z + t_y/t_z}{-r_{31}d/t_z + 1} = \frac{f}{v_1^c},$$

$$\frac{-r_{12}d/t_z + t_x/t_z}{-r_{32}d/t_z + 1} = \frac{u_2^c}{v_2^c}, \quad \frac{-r_{22}d/t_z + t_y/t_z}{-r_{32}d/t_z + 1} = \frac{f}{v_2^c}, \tag{9}$$

$$\frac{r_{11}d/t_z + t_x/t_z}{r_{31}d/t_z + 1} = \frac{u_3^c}{v_3^c}, \quad \frac{r_{21}d/t_z + t_y/t_z}{r_{31}d/t_z + 1} = \frac{f}{v_3^c}.$$

For ease of expression, we make the following definitions:

$$u_i^c = m_i v_i^c, \quad f = n_i v_i^c, \quad i = 1, 2, 3, \tag{10}$$

$$a_1 = t_x/t_z, \quad a_2 = t_y/t_z, \quad a_3 = r_{11}/t_z, \quad a_4 = r_{21}/t_z,$$
$$a_5 = r_{31}/t_z, \quad a_6 = r_{12}/t_z, \quad a_7 = r_{22}/t_z, \quad a_8 = r_{32}/t_z. \tag{11}$$

Substituting (10) and (11) into (9) gives

$$\frac{-da_3 + a_1}{-da_5 + 1} = m_1, \quad \frac{-da_4 + a_2}{-da_5 + 1} = n_1,$$

$$\frac{-da_6 + a_1}{-da_8 + 1} = m_2, \quad \frac{-da_7 + a_2}{-da_8 + 1} = n_2, \tag{12}$$

$$\frac{da_3 + a_1}{da_5 + 1} = m_3, \quad \frac{da_4 + a_2}{da_5 + 1} = n_3.$$

In (12), only $a_i(i = 1, 2, ..., 8)$ are unknown quantities, which can be simplified as

$$
\begin{aligned}
a_1 &= M_2 d^2 a_5 + M_1, \quad a_2 = N_2 d^2 a_5 + N_1, \\
a_3 &= M_1 a_5 + M_2, \quad a_4 = N_1 a_5 + N_2, \\
a_6 &= m_2 a_8 - M_2 d a_5 + M_3, \quad a_7 = n_2 a_8 - N_2 d a_5 + N_3,
\end{aligned}
\tag{13}
$$

where

$$
M_1 = \frac{m_1 + m_3}{2}, \quad N_1 = \frac{n_1 + n_3}{2},
$$

$$
M_2 = \frac{m_1 - m_3}{2d}, \quad N_2 = \frac{n_1 - n_3}{2d},
\tag{14}
$$

$$
M_3 = \frac{2m_2 - m_1 - m_3}{2d}, \quad N_3 = \frac{2n_2 - n_1 - n_3}{2d}.
$$

By the nature of the rotation matrix, we have

$$
r_{11} r_{12} + r_{21} r_{22} + r_{31} r_{32} = 0,
\tag{15}
$$

$$
r_{11}^2 + r_{21}^2 + r_{31}^2 = r_{12}^2 + r_{22}^2 + r_{32}^2 = 1.
\tag{16}
$$

Divide both sides of (15) and (16) by $t_z^2$, and substitute (11) and (13) into, and we can obtain

$$
p_1 a_5^2 + p_2 a_5 a_8 + p_3 a_5 + p_4 a_8 + p_5 = 0,
\tag{17}
$$

$$
q_1 a_8^2 + q_2 a_5^2 + q_3 a_5 a_8 + q_4 a_8 + q_5 a_5 + q_6 = 0.
\tag{18}
$$

where

$$
\begin{aligned}
p_1 &= -d(M_1 M_2 + N_1 N_2), \\
p_2 &= m_2 M_1 + n_2 N_1 + 1, \\
p_3 &= M_1 M_3 + N_1 N_3 - d(M_2^2 + N_2^2), \\
p_4 &= m_2 M_2 + n_2 N_2, \\
p_5 &= M_2 M_3 + N_2 N_3,
\end{aligned}
\tag{19}
$$

$$
\begin{aligned}
q_1 &= m_2^2 + n_2^2 + 1, \\
q_2 &= d^2(M_2^2 + N_2^2) - M_1^2 - N_1^2 - 1, \\
q_3 &= -2d(m_2 M_2 + n_2 N_2), \\
q_4 &= 2m_2 M_3 + n_2 N_3, \\
q_5 &= -2d(M_2 M_3 + N_2 N_3) - 2(M_1 M_2 + N_1 N_2), \\
q_6 &= M_3^2 + N_3^2 - M_2^2 - N_2^2.
\end{aligned}
\tag{20}
$$

From (17) we can also obtain

$$
a_8 = -\frac{p_1 a_5^2 + p_3 a_5 + p_5}{p_2 a_5 + p_4}.
\tag{21}
$$

By substituting (21) into (18) and simplifying it, we can obtain

$$
s_1 a_5^4 + s_2 a_5^3 + s_3 a_5^2 + s_4 a_5 + s_5 = 0,
\tag{22}
$$

where

$$s_1 = p_1^2 q_1 + p_2^2 q_2 - p_1 p_2 q_3,$$

$$s_2 = 2 p_1 p_3 q_1 + 2 p_2 p_4 q_2 - p_1 p_4 q_3 - p_2 p_3 q_3 - p_1 p_2 q_4 + p_2^2 q_5,$$

$$s_3 = p_3^2 q_1 + 2 p_1 p_5 q_1 + p_4^2 q_2 - p_3 p_4 q_3 - p_2 p_5 q_3 - p_1 p_4 q_4 + p_2^2 q_6 - p_2 p_3 q_4 + 2 p_2 p_4 q_5, \quad (23)$$

$$s_4 = 2 p_3 p_5 q_1 - p_4 p_5 q_3 - p_3 p_4 q_4 - p_2 p_5 q_4 + p_4^2 q_5 + p_2 p_4 q_6,$$

$$s_5 = p_5^2 q_1 - p_4 p_5 q_4 + p_4^2 q_6.$$

Using the formula for the roots of an unary quartic equation, we can quickly obtain the value of $a_5$ by (22). The filtering of multiple solutions is described in the next subsection. The remaining value of $a_i$ can then be solved for by (13) and (21).

From (11) and (16), we can obtain the value of $t_z$ by

$$t_z = \frac{1}{\sqrt{a_3{}^2 + a_4{}^2 + a_5{}^2}}, \quad (24)$$

and solve for the values of $t_x$ and $t_y$ from (11). Here, we use the non-negativity of $t_y$ to exclude the wrong solution of (24) and obtain the translation vector $\boldsymbol{t}_c^m$. Since rotation matrices are special orthogonal matrices, $\boldsymbol{R}_c^m$ also satisfies

$$r_{ij} = A_{ij}, \quad i, j = 1, 2, 3, \quad (25)$$

where $A_{ij}$ stands for the algebraic cosine formula of $r_{ij}$. So, the rotation matrix $\boldsymbol{R}_c^m$ can be solved from (11) and (25). Due to the accuracy limitations of the actual calculations, Schmidt orthogonalization of $\boldsymbol{R}_c^m$ is also required.

*4.3. Conversion of Coordinate Systems*

The relative localization model of the two UAVs is shown in Figure 4. Multiple coordinate systems are established with $O_b$, $O_c$, and $O_m$ as the origin, respectively. The definitions of $O_c$ and $O_m$ are given in the previous section, and $O_b$ is determined in the same way as $O_m$. $O_{bi}x_{bi}y_{bi}z_{bi}$, $O_{ci}x_{ci}y_{ci}z_{ci}$, and $O_{ui}x_{ui}y_{ui}z_{ui}$ are three inertial coordinate systems, so each of their axes corresponds to parallel, respectively. $O_c x_c y_c z_c$ and $O_m x_m y_m z_m$ are defined in the previous section. $O_b x_b y_b z_b$ and $O_u x_u y_u z_u$ are the fuselage coordinate systems of the two UAVs, where the $x_b(x_u)$-axis points directly to the right of the fuselage, the $y_b(y_u)$-axis points directly in front, and the $z_b(z_u)$-axis is perpendicular to $O_b x_b y_b (O_u x_u y_u)$ and points above the fuselage. The difference between $O_u x_u y_u z_u$ and $O_m x_m y_m z_m$ is that unlike $O_m x_m y_m z_m$, which is set up to simplify calculations, $O_u x_u y_u z_u$ is a common coordinate system used when expressing UAV attitude. Due to the symmetry of the quadcopter UAV, we start by assuming that the positive direction of the $y_u$-axis is always in the first quadrant of the $O_m x_m y_m$.

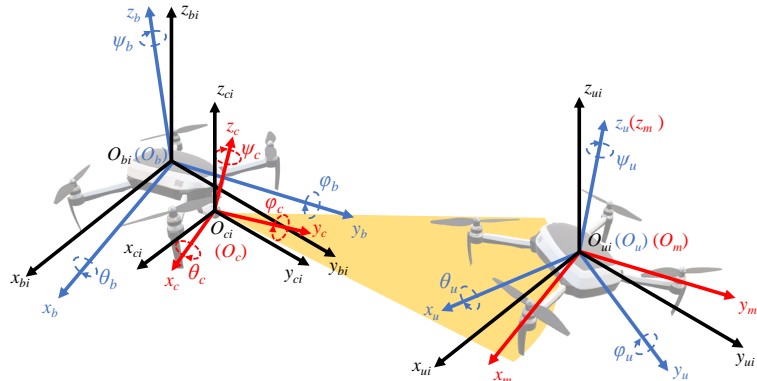

**Figure 4.** The coordinate system of interest for relative localization of the UAV.

Obviously, the relative position of the positioned UAV can be expressed as $t_{bi}^u = O_{bi}O_u$. Due to the same orientation of the inertial coordinate systems, the attitude of the positioned UAV can be expressed as the rotation matrix $R_{bi}^u$ of $O_u x_u y_u z_u$ with respect to $O_b x_b y_b z_b$. $R_{bi}^u$ and $t_{bi}^u$ can be considered as the result of a series of coordinate system transformations and the flexible kinematic properties of UAVs and gimbals increase the difficulty of solving them.

The solution scheme for $R_c^m$ and $t_c^m$ is given in the previous section. The attitude rotation matrices of the localization UAV and gimbal can be obtained based on their Euler angles acquired in real time. The Euler angle consists of roll angle $\varphi$, pitch angle $\theta$, and yaw angle $\psi$, and the order of rotation is, based on an inertial coordinate system, first $\psi$ degrees around the $z$-axis, then $\theta$ degrees around the transformed $x$-axis, and finally $\varphi$ degrees around the transformed $y$-axis. The conversion formulas for Euler angles to the rotation matrix $R$ in the right-handed coordinate system are

$$R_x(\theta) = \begin{bmatrix} 1 & 0 & 0 \\ 0 & \cos\theta & -\sin\theta \\ 0 & \sin\theta & \cos\theta \end{bmatrix},$$

$$R_y(\varphi) = \begin{bmatrix} \cos\varphi & 0 & \sin\varphi \\ 0 & 1 & 0 \\ -\sin\varphi & 0 & \cos\varphi \end{bmatrix}, \tag{26}$$

$$R_z(\psi) = \begin{bmatrix} \cos\psi & -\sin\psi & 0 \\ \sin\psi & \cos\psi & 0 \\ 0 & 0 & 1 \end{bmatrix},$$

and

$$R = R_z(\psi) \cdot R_x(\theta) \cdot R_y(\varphi). \tag{27}$$

The attitude rotation matrices $R_{bi}^b$ and $R_{ci}^c$ can be obtained by substituting the Euler angles $\varphi_b, \theta_b, \psi_b$ and $\varphi_c, \theta_c, \psi_c$ of the localization UAV and the gimbal into (26) and (27), respectively.

Based on the above known information, we give the solution scheme for $R_{bi}^u$ and $t_{bi}^u$. Since the isotropy of inertial coordinate systems it follows that

$$R_{bi}^u = R_{ci}^u. \tag{28}$$

where $R_{ci}^u$ denotes the rotation matrix of the positioned UAV relative to the camera inertial coordinate system. By the transitivity of the rotation matrix, $R_{ci}^u$ can be expressed as

$$R_{ci}^u = R_{ci}^c \cdot R_c^m \cdot R_m^u, \tag{29}$$

where, according to the direction in which the coordinate system is set up, it is easy to know that

$$R_m^u = R_z(-\frac{\pi}{4}). \tag{30}$$

By the additive property of vectors, $t_{bi}^u$ can be expressed as

$$t_{bi}^u = t_{bi}^{ci} + t_{ci}^u, \tag{31}$$

where $t_{bi}^{ci}$ can be obtained from

$$t_{bi}^{ci} = R_{bi}^b \cdot t_0, \tag{32}$$

where $t_0$ represents the initial value of $t_{bi}^{ci}$ when $\varphi_b, \theta_b, \psi_b = 0$, which can be easily obtained by measurement. And we can obtain $t_{ci}^u$ by

$$t_{ci}^u = R_{ci}^c \cdot t_c^m. \tag{33}$$

In summary, the relative position and attitude of the positioned UAV are finally given as

$$t^u_{bi} = R^b_{bi} \cdot t_0 + R^c_{ci} \cdot t^m_c,$$
$$R^u_{bi} = R^c_{ci} \cdot R^m_c \cdot R^u_m. \tag{34}$$

### 4.4. Determination of Correct Solution

Theoretically, the quartic equation of one unknown of (22) has at most four different real roots. However, according to the conclusions of [47], in the P3P problem, the equation can be considered to have only two sets of real solutions, i.e., two sets of three-dimensional spatial points can be derived from one set of two-dimensional projected points. We verified this conclusion in simulation experiments, and the simulation model is shown in Section 5.

The two sets of solutions correspond to two sets of UAV positions and attitudes, as shown in Figure 5. $\{M'_i\}(i = 1, 2, 3, 4)$ represents another set of erroneous motor positions derived from the projected points $\{P_i\}$, and $O'_m$ is the erroneous position of the UAV. The degree of inclination of the UAV body corresponding to the two sets of solutions can be represented by the angle $\angle z_u O_m z_{ui}$ and angle $\angle z'_u O'_m z'_{ui}$, which are set as $\beta_u$ and $\beta'_u$, respectively.

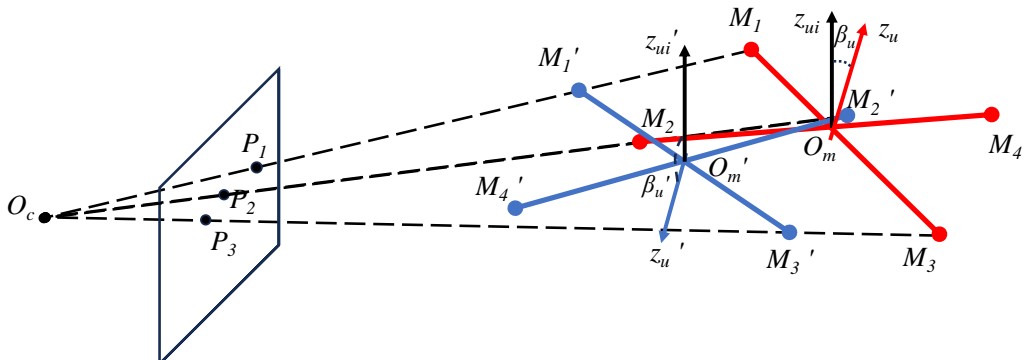

**Figure 5.** The position and attitude of the UAV corresponding to the two sets of solutions.

$\beta_u$ is a result of the roll and pitch that occurs in the UAV, so the value of $\beta_u$ should be within a limited range during normal flight. According to the vector angle formula, we can obtain

$$\cos \beta_u = \frac{w_3 \cdot z_{ci}}{\|w_3\|\|z_{ci}\|}, \tag{35}$$

where $w_3$ denotes the third row of $R^u_{bi}$, which also represents the unit vector of the $z_u$-axis in the inertial coordinate system. Let $w_3 = [w_{31}, w_{32}, w_{33}]$ and $z_{bi} = [0, 0, 1]$; $\beta_u$ can be obtained from

$$\beta_u = \arccos w_{33}. \tag{36}$$

From (26) and (27), we have $w_{33} = \cos \varphi_u \cos \theta_u$. The roll and pitch angles of UAVs are usually finite, denoted as $\theta_u \in [\varphi^{min}_u, \varphi^{max}_u]$ and $\theta_u \in [\theta^{min}_u, \theta^{max}_u]$. And, due to the symmetry of quadrotor UAVs, usually $\varphi^{max}_u = \theta^{max}_u = -\varphi^{min}_u = -\theta^{min}_u$. Then, the range of $\beta_u$ can be expressed as

$$\beta_u \in [\, 0, \cos^2 \varphi^{max}_u]. \tag{37}$$

We therefore set the maximum value of pitch and roll angles uniformly to $\alpha^{max}_u$.

Since it is difficult to obtain the range of $\beta'_u$ by mathematical derivation, we each obtained the approximate distribution of $\beta'_u$ at $\varphi^{max}_u = \theta^{max}_u = \pi/6$ and $\varphi^{max}_u = \theta^{max}_u = \pi/4$ based on 10,000 simulation experiments, respectively, as shown in Figure 6.

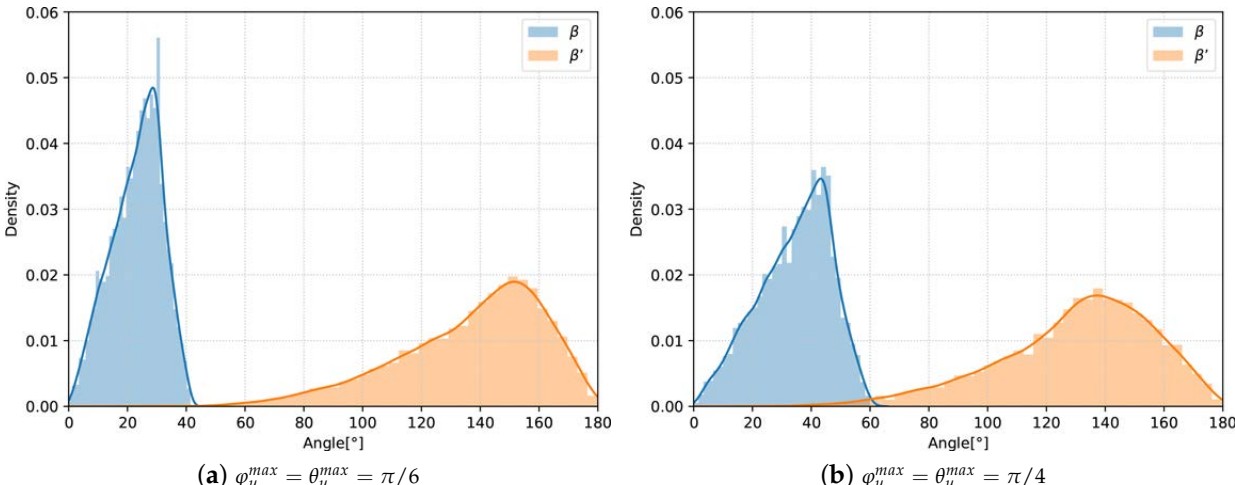

**(a)** $\varphi_u^{max} = \theta_u^{max} = \pi/6$  **(b)** $\varphi_u^{max} = \theta_u^{max} = \pi/4$

**Figure 6.** Distribution of UAV body tilt angles corresponding to the two sets of solutions.

It can be seen that the vast majority of the values of $\beta_u'$ are greater than $\beta_u^{max}$, the maximum value of $\beta_u$, compared to the values of $\beta_u$ that are strictly in the range shown in (37). In the two sets of experiments, the values of $\beta_u'$ greater than $\beta_u^{max}$ are approximately 99.8% and 98.8%, respectively. Therefore, in the vast majority of cases, the correct solution can be identified based on the value of $\beta_u^{max}$. Subject to errors in the projection points of motors, the value of $\beta_u^{max}$ tends to be slightly larger than $\cos^2 \varphi_u^{max}$. Approximate values of $\beta_u^{max}$ can be obtained based on a large number of simulation experiments.

When $\beta_u'$ is also smaller than $\beta_u^{max}$, partially incorrect solutions can be further detected based on whether $\theta_u$ and $\varphi_u$ corresponding to each set of solutions are simultaneously smaller than $\varphi_u^{max}$ and $\theta_u^{max}$, respectively. We set the maximum value of pitch and roll angles uniformly to $\alpha_u^{max}$. Similar to $\beta_u^{max}$, the actual values obtained for $\alpha_u^{max}$ are slightly larger than $\varphi_u^{max}$ and $\theta_u^{max}$, and their approximations can be obtained through extensive randomized experiments.

For the mis-solutions that remain unfiltered, we find that their average error is much smaller than the measured distance and much lower than the average error of the full set of mis-solutions. When $\varphi_u^{max} = \theta_u^{max} = \pi/6$ and $\varphi_u^{max} = \theta_u^{max} = \pi/4$, simulation results show that the average errors of these incorrect solutions are only about $0.05\%\|t_{bi}^u\|$ and $0.63\%\|t_{bi}^u\|$, which are about 1/30 and 2/5 of the overall average error, respectively. We therefore take the average of these group solution pairs as the result.

In summary, the algorithm for determining the correct solution is shown in Algorithm 2:

### 4.5. Four Motors Detected

When all four motors are detected, positioning accuracy can be further improved. We divide the four projection points of motors into groups of three each in the order specified in Section 3.2. By substituting each of the four sets of projection points into the above solution scheme, four sets of localization results can be obtained. We set $t_i$ to denote the relative position obtained based on the three points other than point $P_i$.

The keypoint detection module gives the detection confidence for each motor, set to $c_{1:4}$. The weight $W_i$ of $t_i$ can be obtained based on $c_i$ by

$$W_i = \frac{(\sum\limits_{j=1}^{4} c_j) - c_i}{3 \sum\limits_{j=1}^{4} c_j}. \tag{38}$$

Then $t_{bi}^u$ can be given by

$$t_{bi}^u = \sum_{i=1}^{4} W_i t_i. \tag{39}$$

---

**Algorithm 2** Determining the correct solution

---

**Require:** $T = \{t_{bi1}^u, t_{bi2}^u\}$, $B = \{\beta_{u1}, \beta_{u2}\}$, $A = \{\{\theta_{u1}, \varphi_{u1}\}, \{\theta_{u2}, \varphi_{u2}\}\}$
**Ensure:** $t_{bi}^u$

1: **if** $min(B) < \beta_u^{max}$ and $max(B) > \beta_u^{max}$ **then**
2:      $idx = min(B)$'s index of $B$
3: **else if** $max(abs(A_1)) < \alpha_u^{max}$ and $max(abs(A_2)) > \alpha_u^{max}$ **then**
4:      $idx = 1$
5: **else if** $max(abs(A_1)) > \alpha_u^{max}$ and $max(abs(A_2)) < \alpha_u^{max}$ **then**
6:      $idx = 2$
7: **else if** $max(abs(A_1)) < \alpha_u^{max}$ and $max(abs(A_2)) < \alpha_u^{max}$ **then**
8:      $idx = 0$
9: **end if**
10: **if** $idx = 0$ **then**
11:      $t_{bi}^u = \frac{t_{bi1}^u + t_{bi2}^u}{2}$
12: **else**
13:      $t_{bi}^u = T_{idx}$
14: **end if**

---

### 4.6. Two Motors Detected

Since the case where only two motors are detected rarely occurs, we give a transitional estimation scheme. The problem model at this point is shown in Figure 7.

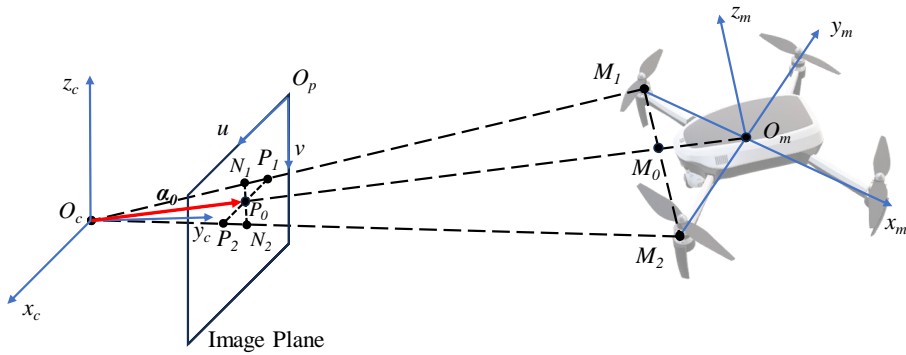

**Figure 7.** Schematic diagram when two motors are detected.

Taking into account the occlusion, we approximate that $O_c$ is coplanar with $\{M_{1:4}\}$ and that $\|O_c M_1\| = \|O_c M_2\|$. So, $O_c O_m$ intersects $M_1 M_2$ at the midpoint of $M_1 M_2$ and the intersection is set to $M_0$. The projection point of $O_m$ on the image plane is set to $P_0$ and $\alpha_0$ represents the vector $O_c P_0$. Then, the displacement vector $t_c^m$ can be expressed as

$$t_c^m = \frac{\|O_c M_0\| + \|O_m M_0\|}{\|\alpha_0\|} \alpha_0, \tag{40}$$

where $\|O_m M_0\|$ is known to be $\frac{\sqrt{2}}{2}d$.

Make a parallel line of $M_1M_2$ through $P_0$, intersecting $O_cM_1$ and $O_cM_2$ at $N_1$ and $N_2$, respectively. From the properties of similar triangles we have

$$\frac{\|\boldsymbol{\alpha_0}\|}{\|\boldsymbol{O_cM_0}\|} = \frac{\|\boldsymbol{N_1N_2}\|}{\|\boldsymbol{M_1M_2}\|}, \tag{41}$$

where it is easy to see that $\|\boldsymbol{M_1M_2}\| = \sqrt{2}d$. Since $P_1$ and $P_2$ are known, the angles of $\angle P_1O_cP_2$, $\angle O_cP_1P_2$, and $\angle O_cP_2P_1$ can be obtained based on the vector pinch equations, which are set to $\eta_1$, $\eta_2$ and $\eta_3$, respectively. Here, it is specified that $\eta_2 < \pi/2 < \eta_3$. By the sine theorem, it can be obtained that

$$\begin{aligned} \frac{\|\boldsymbol{P_0N_1}\|}{\sin \eta_2} &= \frac{\|\boldsymbol{P_0P_1}\|}{\sin(\frac{\pi}{2} + \frac{\eta_1}{2})}, \\ \frac{\|\boldsymbol{P_0N_2}\|}{\sin(\pi - \eta_3)} &= \frac{\|\boldsymbol{P_0P_2}\|}{\sin(\frac{\pi}{2} - \frac{\eta_1}{2})}. \end{aligned} \tag{42}$$

It is also known that

$$\|\boldsymbol{P_0N_1}\| = \|\boldsymbol{P_0N_2}\|, \tag{43}$$

and

$$\|\boldsymbol{P_0P_1}\| + \|\boldsymbol{P_0P_2}\| = \|\boldsymbol{P_1P_2}\|. \tag{44}$$

From (42)–(44), we can obtain

$$\|\boldsymbol{N_1N_2}\| = 2\|\boldsymbol{P_1P_2}\| \frac{\sin \eta_2 \sin(\pi - \eta_3)}{\sin(\frac{\pi}{2} + \frac{\eta_1}{2}) \sin(\pi - \eta_3) + \sin \eta_2 \sin(\frac{\pi}{2} - \frac{\eta_1}{2})}, \tag{45}$$

and

$$\|\boldsymbol{\alpha_0}\| = \frac{\frac{1}{2}\|\boldsymbol{N_1N_2}\|}{\tan \frac{\eta_1}{2}}. \tag{46}$$

Then, we can obtain $\|\boldsymbol{O_cM_0}\|$ first by (41) and then $t_c^m$ by (40). Finally, after the coordinate transformation of Section 4.3, $t_{bi}^u$ can be obtained.

## 5. Experimental Results and Analysis

Our experiment is divided into three parts. First, we obtained a self-training model of YOLOv8 by training based on the captured images and tested its effectiveness in detecting experimental UAVs and their motors. In the second part, we constructed the high-fidelity airborne gimbal camera model and localized UAV model based on the actual parameters, and examined the performance of the relative localization algorithm in various situations. Finally, we conducted system experiments based on two UAVs to verify the feasibility of our overall scheme using GPS-based relative localization data as a reference.

### 5.1. Experiment Platform

The hardware composition and operational architecture of the UAV experimental platform used to validate the proposed scheme is shown in Figure 8. We conduct secondary development and experiments based on two *Prometheus* 450 (*P450*) UAVs produced by *Amovlab*, Chengdu, China [48]. Each UAV is equipped with NVIDIA's Edge AI supercomputer Jeston Xavier NX and a Pixhawk 4 flight controller. The Jeston Xavier NX has a hexa-core NVIDIA Carmel ARM CPU, 6GB of LPDDR4x RAM and a GPU with 21TOPS of AI inference performance, which is capable of meeting the arithmetic requirements under Ubuntu 18.04. The Pixhawk 4 flight controller is the control hub of the UAV. We retrofitted the UAV with *amovlab*'s G1 gimbal camera to stream real-time images to the Jeston Xavier NX. The edge computer also obtains attitude data from the gimbal and flight controller through their ROS topics published in real time via the serial port. Based on the above data, the UAV achieves real-time detection and relative localization for other UAVs within

its visual perception range on the Jeston Xavier NX. All experimental data were obtained based on this platform system. Key parameters of the UAV: $d = 21$ cm, $t_0 = [0, 13, -6]$ cm.

### 5.2. Detection Performance Experiment

We labeled 1250 collected images of experimental UAVs and used them as a dataset to obtain a self-training model by training. We conducted UAV-to-UAV target detection experiments at distances ranging from 2 to 12 m. The experimental results show that the YOLOv8-pose target detection module based on the self-trained model is able to stably detect the target UAV and its visible motors. The motor's image plane positioning point can basically remain within the range of the motor's projected image. Screenshots of the detection results are shown in Figure 9, where the motors are marked by blue dots. The average detection time of the on-board target detection module for each image frame is about 43.5 ms.

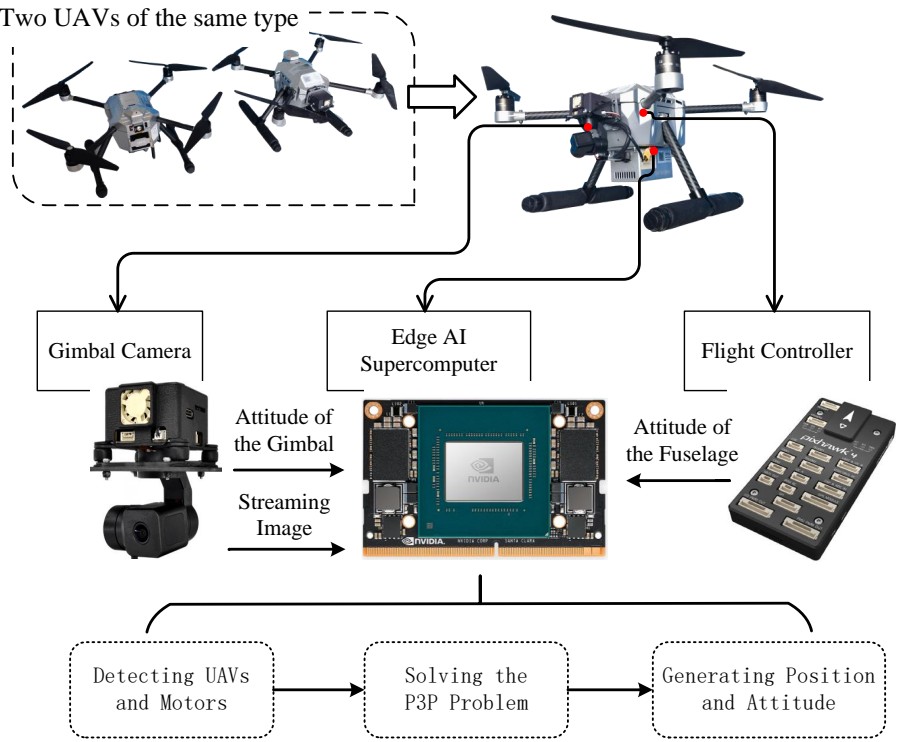

**Figure 8.** The hardware composition and operational architecture of the UAV experimental platform.

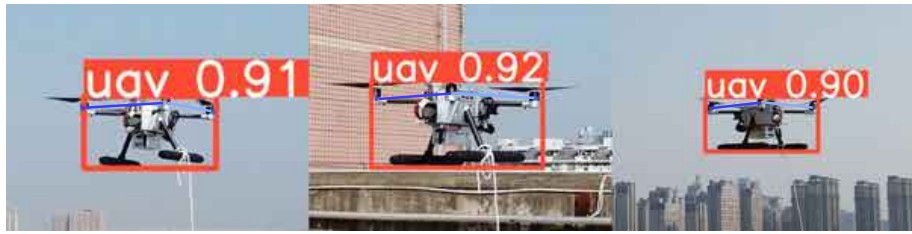

**Figure 9.** Detection effects of the UAV and its motors.

In summary, we verified the feasibility of realizing real-time detection of UAVs and their motors with an airborne camera based on YOLOv8.

### 5.3. Relative Localization Simulation Experiment

We tested the speed and accuracy of the proposed algorithm based on a self-built simulation model and compared it with three mainstream algorithms, which are Gao's, iterative method (IM) and AP3P. In order to increase the fidelity, all of our simulation experiments were performed on the edge computer of the *P*450 UAV.

5.3.1. Simulation Model

We constructed a virtual camera model based on the parameters of the *G1* gimbal camera with an intrinsic matrix $K$ of

$$K = \begin{bmatrix} 640 & 0 & 640 \\ 0 & 640 & 360 \\ 0 & 0 & 1 \end{bmatrix}. \tag{47}$$

Based on the camera calibration work that has been performed, we assume that the camera's distortion is zero. The pitch angle of the gimbal $\theta_c \in [-\pi/3, \pi/3]$. The camera is capable of detecting drones from 2 to 12 m away from itself, which means that $D \in [2, 12]$ m, where $D = \|t_{true}\|$.

In order to describe the situation where the motor is obscured, we designed a UAV model based on the *P450*, as shown in Figure 10. In the aforementioned $O_m x_m y_m z_m$ coordinate system, the body of the fuselage is represented by a sphere with $O_f$ as the center and radius $R = 10$ cm, and the motors are represented by spheres with $M_{1:4}$ as the center and radius $r = 2$ cm. The coordinate of $O_f$ is $[\, 0,\ 0,\ -5\,]$ cm.

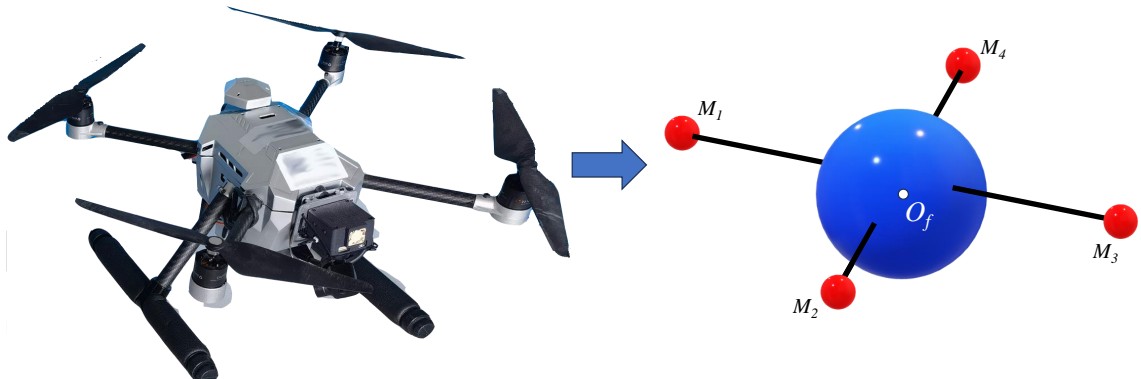

**Figure 10.** Simplification of the UAV.

The attitude of the UAV is determined by randomly generated Euler angles and Euler angles $\varphi_b, \theta_b, \psi_b \in [-\pi/4, \pi/4]$. The coordinates of $O_f$ and $M_{1:4}$ in the Oxyz coordinate system can be obtained based on the Euler angles. Then, based on the projection relation, the projection points $P_f$ and $P_{1:4}$ of $O_f$ and $M_{1:4}$ on the image plane, and the radius $R_p$ and $r_{p1:p4}$ of the projection circles of the fuselage and motors can be obtained.

According to the masking relation, the decision condition that three motors can be detected is expressed as

$$\|P_f P_4\| < R_p, \tag{48}$$

and the decision condition for detecting only two motors is

$$\|P_1 P_4\| < r_{p1} \wedge \|P_2 P_3\| < r_{p2}. \tag{49}$$

To simulate the error in motor detection, we add white noise obeying a two-dimensional Gaussian distribution to the image plane projection point $\{P_i(u_i, v_i)\}\ (i = 1, 2, 3, 4)$ of motors, i.e., the actual projection point $P_i'(u_i', v_i')$ is denoted as

$$(u_i',\ v_i') \sim N(u_i,\ v_i,\ \sigma_{i1}^2,\ \sigma_{i2}^2,\ 0), \tag{50}$$

where

$$\sigma_{i1} = \sigma_{i2} = \sigma \frac{f}{y_i}. \tag{51}$$

$\sigma$ is the standard deviation in centimeters of the 3D spatial point corresponding to the motor's localization point on the image plane and the position of the motor's true point.

$f$ represents the focal length and $y_i$ denotes the coordinates of the motor $M_i$ in the $y$-axis under the camera coordinate system, in meters.

We designed three values of $\sigma$, which are 0.5 cm, 1.0 cm, and 1.5 cm, based on the actual radius of the $P450$, which is 2 cm for the motor. The three values from small to large correspond to high to low accuracy and can be described as the localization point basically on the motor center, basically on the motor, and partially on the motor, respectively.

### 5.3.2. Execution Speed

The time taken to solve the P3P problem is the main factor affecting the speed of the relative localization algorithm. We performed execution time tests of the proposed algorithm as well as other classical algorithms at the same performance state of the edge computer. Each algorithm was run for 10,000 rounds. The distribution of single execution time is shown in Figure 11, and the average time taken is shown in Table 2.

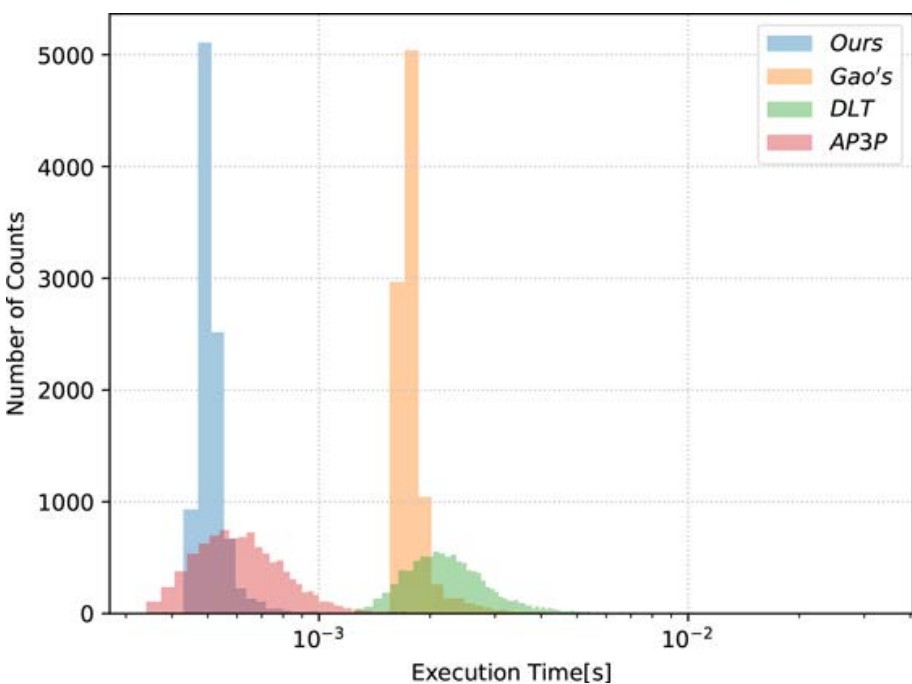

**Figure 11.** Distribution of single execution time for four algorithms.

**Table 2.** Average single execution time for the four algorithms.

| Algorithms | Time [ms] | Proportionality |
|:---:|:---:|:---:|
| Ours | 0.534 | 1 |
| Gao's | 1.845 | 3.46 |
| IM | 2.614 | 4.90 |
| AP3P | 0.722 | 1.35 |

It can be seen that our algorithm executes approximately 3.5 times faster than Gao's, 5 times faster than IM, and 35% faster than AP3P. Experimental results show that our proposed algorithm executes significantly faster than Gao's and IM. Compared to AP3P, we have a smaller but more consistent speed advantage. This is largely due to the fact that we have taken full advantage of the geometric characteristics of UAVs for targeted problem modeling. Our algorithm takes relative position as the unique objective and solves for it directly instead of obtaining it indirectly, reducing the accumulation and amplification of errors. Based on the results of the previous mathematical derivation, we only need to carry out simple algebraic calculations in the actual solution, which avoids the solution of the angle and the operation of the matrix and significantly reduces the computational complexity.

5.3.3. Computational Accuracy

In order to measure the accuracy of the relative localization and the correct choice of the solution, we denote the relative localization error as

$$e_t = \| \boldsymbol{t}_{est} - \boldsymbol{t}_{true} \|. \tag{52}$$

Following the approach of Section 4.4, we obtain reasonable values of $\beta_u^{max}$ and $\alpha_u^{max}$ for three levels of detection accuracies with a sufficient number of randomized simulation experiments with known correct solutions. The values taken are shown in Table 3.

**Table 3.** Values of $\beta_u^{max}$ and $\alpha_u^{max}$ for different detection accuracies.

| $\sigma$ [cm] | $\beta_u^{max}$ | $\alpha_u^{max}$ |
|---|---|---|
| 0.5 | 70° | 52° |
| 1.0 | 75° | 58° |
| 1.5 | 80° | 62° |

We randomly generated 10,000 sets of UAV position and attitude data in the simulation scenario. According to our occlusion model determination, there are 7871 sets of data where all four motors are detected, 2114 sets of data where three motors are detected, and 15 sets of data where only two motors are detected. This suggests that it is common for all four motors not to be detected. And given the simplified nature of the model and the fact that UAV swarms are often at similar altitudes during actual flight, the probability of detecting less than four motors should be greater. This supports the need for the study.

We first tested the overall accuracy of the proposed algorithm based on the simulation data and the experimental results are shown in Figure 12, and the vertical coordinate indicates the value of the kernel density estimate.

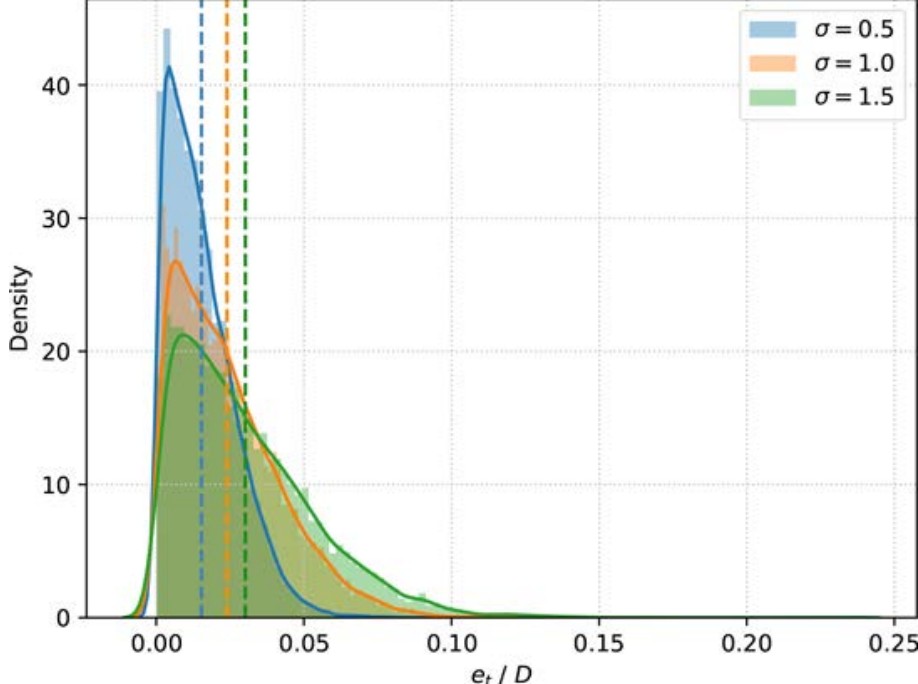

**Figure 12.** Error distributions of our algorithm under three levels of noise corresponding to $\sigma = 0.5$, 1.0 and 1.5, respectively.

The average localization errors at the three levels of noise are 1.53% $D$, 2.39% $D$, and 3.01% $D$, respectively, and are marked with vertical dashed lines in the figure (the same below). The data show that the localization accuracy of our algorithm has generally

stabilized at a high level, and continues to provide less error-prone and stable localization data in the presence of increased noise. To further study the performance of the proposed algorithm, we analyze the specific performance of the algorithm when different numbers of motors are detected.

We solved the 7871 sets of data detected for the four motors by applying Gao's, IM, and AP3P methods, respectively, and compared them with the results of our algorithm. The error distribution of the four algorithms under different levels of noise is shown in Figure 13, and the corresponding average errors are shown in Table 4.

It is clear that the accuracy of IM and AP3P is significantly reduced when noise is present. The large error indicates that these two methods are not applicable to the solution of our research problem. The proposed algorithm is slightly more accurate than Gao's. We speculate that this advantage may stem from our weighting of the data based on the detection confidence of each motor. We speculate that this advantage may be the result of our multi-resolution solution as well as the regrouping weighting process. Therefore, we replaced our proposed post-processing scheme for the P3P solution with the reprojection method used by Gao and compared the experimental results with the results of our and Gao's schemes. The results of this experiment are shown in Figure 14.

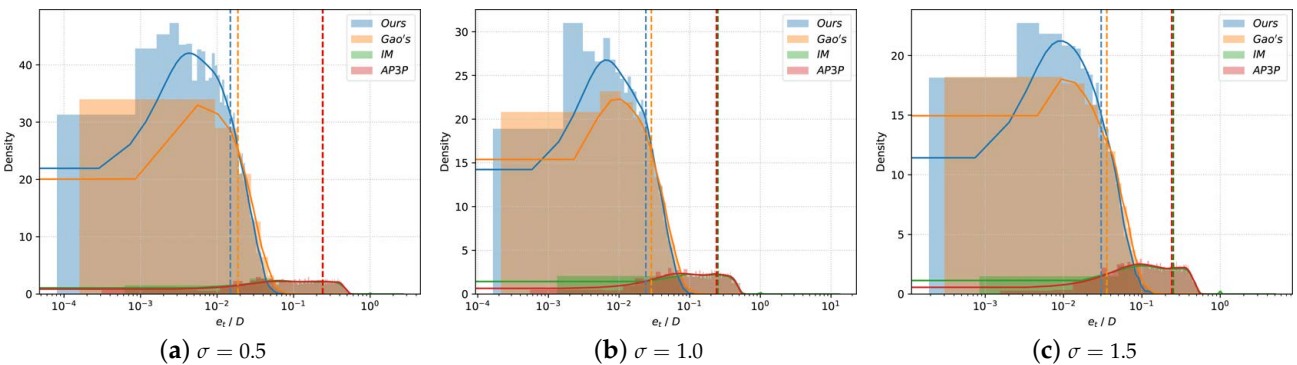

**(a)** $\sigma = 0.5$      **(b)** $\sigma = 1.0$      **(c)** $\sigma = 1.5$

**Figure 13.** Error distributions of the four algorithms for the three noise levels corresponding to $\sigma = 0.5$, 1.0 and 1.5.

**Table 4.** Localization errors of four algorithms with different detection accuracies.

| $\sigma$ [cm] | Ours | Gao's | IM | AP3P |
|---|---|---|---|---|
| 0.5 | 0.015 | 0.019 | 0.242 | 0.239 |
| 1.0 | 0.024 | 0.029 | 0.251 | 0.239 |
| 1.5 | 0.030 | 0.036 | 0.252 | 0.240 |

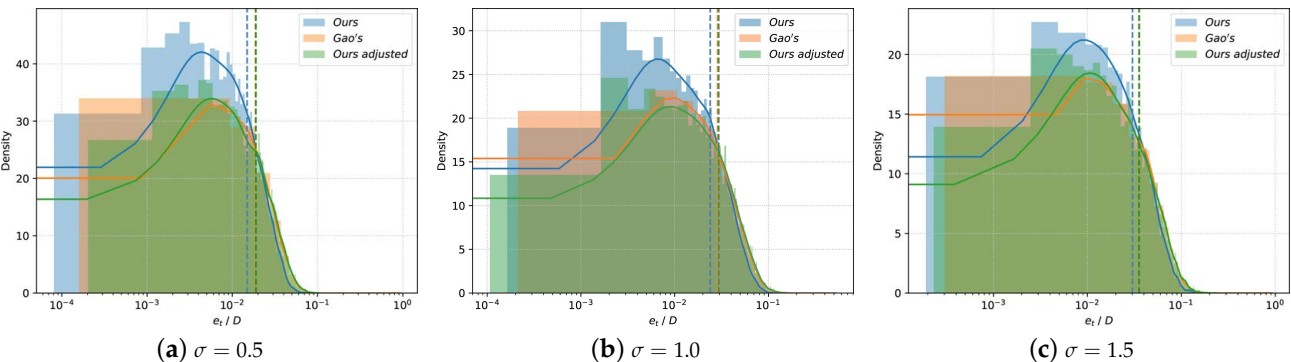

**(a)** $\sigma = 0.5$      **(b)** $\sigma = 1.0$      **(c)** $\sigma = 1.5$

**Figure 14.** Error distributions of our original, adjusted, and Gao's algorithm for three levels of noise corresponding to $\sigma = 0.5$, 1.0, and 1.5.

It can be seen that the accuracy of our algorithm is very close to that of Gao's after using the reprojection method instead of our post-processing scheme. This verifies the effectiveness of our post-processing scheme for accuracy improvement. By comparing the data in detail, we found that our post-processing algorithms are able to keenly detect outliers with large deviations and eliminate them or reduce their impact. Thus, our post-processing algorithm improves the robustness of the solution. However, our regrouping-weighted processing approach increases the computational cost, so we can choose to discard this part of the scheme when the arithmetic power is limited.

Due to the lack of other algorithms for obtaining the correct displacement based on the three key points, we can only compare the localization accuracy when three motors are detected with that when four motors are detected. Additional experiments were conducted, resulting in 7871 sets of localization data based on three motor points at each of the three levels of detection accuracy. The localization errors are shown in Figure 15.

As can be seen from the figure, our algorithm maintains a similar localization accuracy when only three motors are detected as when four motors are detected, specifically 1.68% $D$, 2.58% $D$, and 3.19% $D$. Localization errors still come mainly from detection errors. This shows that the performance of our pose-based multi-resolution determination scheme is robust. In the absence of a fourth motor point as a reprojection point, our method can effectively replace the reprojection method to obtain a stable and accurate solution.

We also tested the performance of the transitional solution when only two motors were detected. We obtained the results of 1,000 sets of experiments through a much larger number of randomized experiments, as shown in Figure 16.

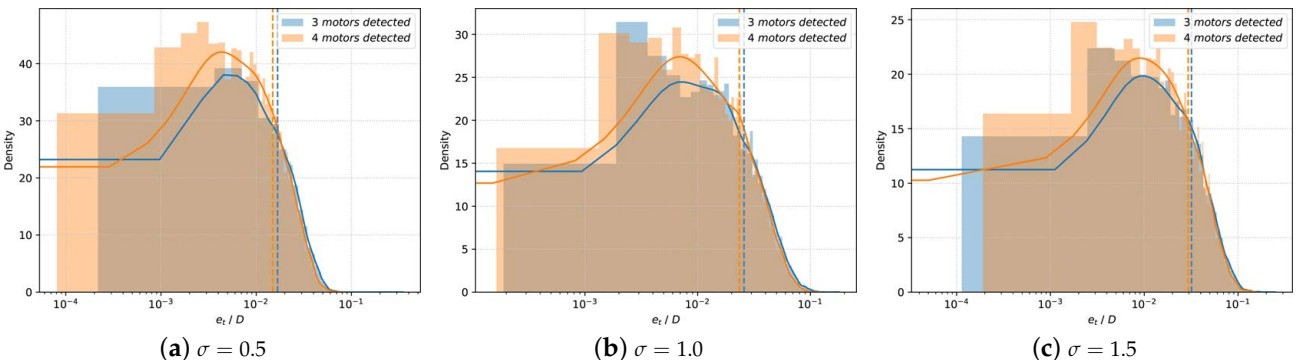

(**a**) $\sigma = 0.5$ (**b**) $\sigma = 1.0$ (**c**) $\sigma = 1.5$

**Figure 15.** Error distribution of our algorithm when only two motors are detected.

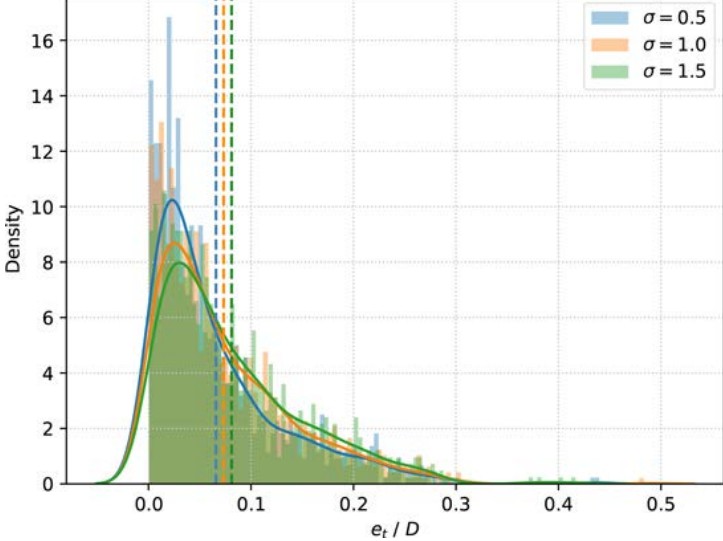

**Figure 16.** The localization error of our algorithm when two motors are detected.

It can be seen that the average error of our localization scheme when detecting two motors is controlled within 10% *D*, specifically 6.58% *D*, 7.33% *D*, and 8.10% *D*, respectively. Although some of the errors are large, given the small probability of the event occurring, we believe that its performance is acceptable as a transitional solution for special cases. In the process of processing data from consecutive frames, it is possible to combine the data from previous frames when more than two motors were detected and reduce the error by methods such as Kalman filtering.

5.3.4. System Experiment

Based on the demonstration of simulation experiments, we conducted real system experiments based on two *P*450 UAVs in a real environment. Due to the temporary lack of other more accurate means of localization, we generate the true relative position coordinates of the two UAVs based on GPS positioning data in an unobstructed environment. To minimize the increase in error due to other factors, we controlled the UAV used for localization to remain hovering in the air, and the localized UAV flew within the field of view of the camera for one minute, as shown in Figure 17. The real-time true relative position during the flight and the estimated relative position based on the proposed algorithm are shown in Figure 18, and Figure 19 illustrates the corresponding error distribution.

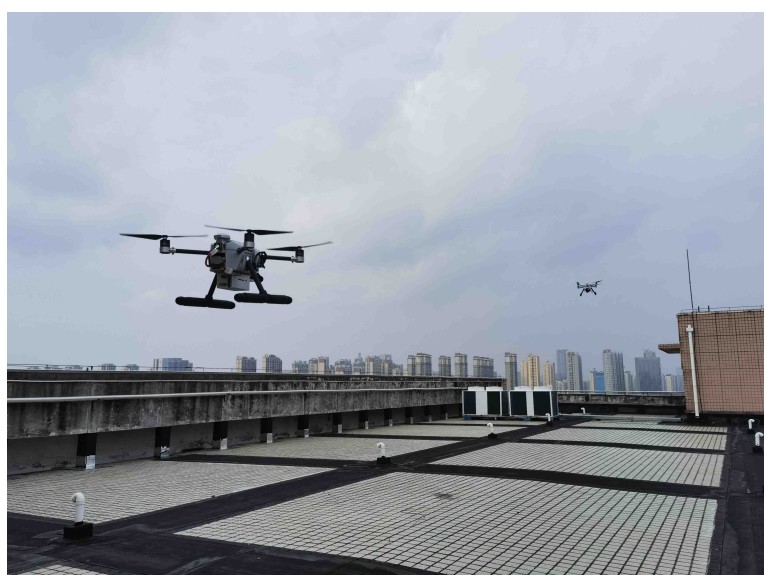

**Figure 17.** Real experimental scene diagram.

As shown in the figures, our scheme is generally able to achieve real-time vision-based relative localization between UAVs. The average relative error of the real experiment is 4.14%, which is slightly larger than the maximum average error of the simulation experiment. The error in the *y*-axis direction is significantly larger than that in the *x*-axis and *z*-axis directions, which is in line with the principle of our scheme. More outliers with larger deviations appear in the estimation results. By analyzing the data, we determined that this was the result of larger errors in the image plane coordinates of the motors. In addition, $t_{true}$ itself, which is generated based on GPS and barometric altimeter data, actually has some error.

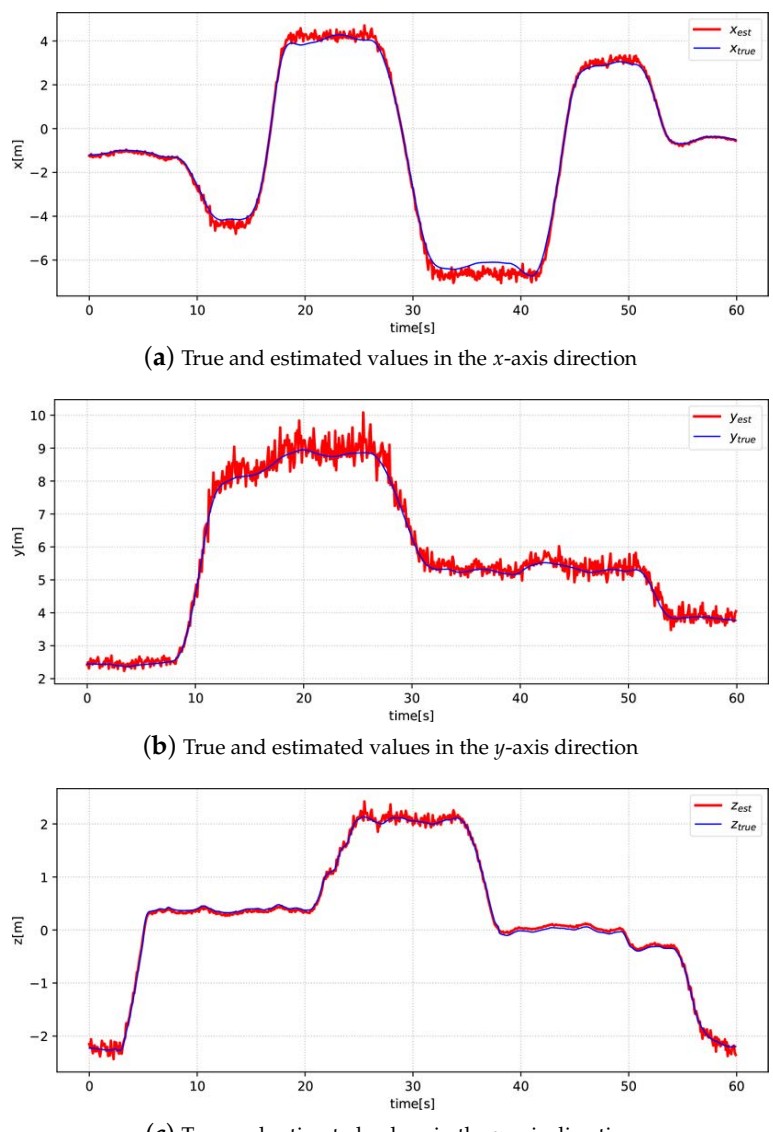

(**a**) True and estimated values in the *x*-axis direction

(**b**) True and estimated values in the *y*-axis direction

(**c**) True and estimated values in the *z*-axis direction

**Figure 18.** Comparison of true and estimated values of relative positions.

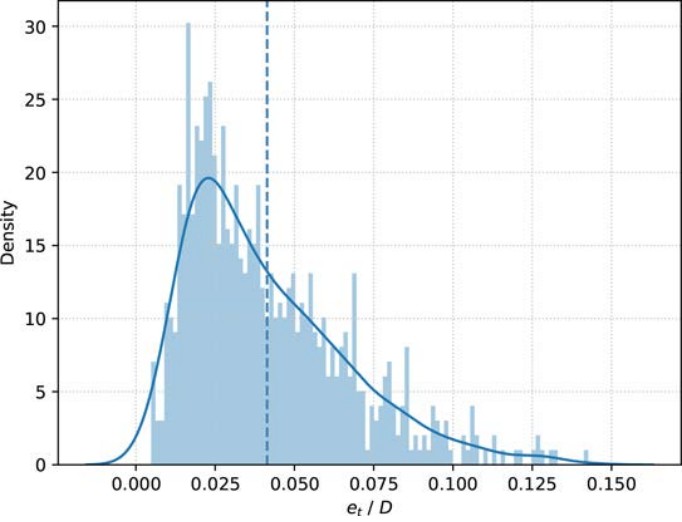

**Figure 19.** Error distribution in real experiments.

## 6. Conclusions

In order to realize real-time accurate relative localization within UAV swarms, we investigate a visual relative localization scheme based on onboard monocular sensing information. The conclusions of the study are as follows:

- Our study validates the feasibility of accurately detecting UAV motors in real time using the YOLOv8-pose attitude detection algorithm.
- Our PnP solution algorithm derived based on the geometric features of the UAV proved to be faster and more stable.
- Through the validation of a large number of stochastic experiments, we propose for the first time a fast scheme based on the rationality of UAV attitude to deal with the PnP multi-solution problem, which ensures the stability of the scheme when the visual information is incomplete.

Our scheme improves speed and accuracy while reducing data requirements, and the performance is verified in experiments.

However, there are limitations to our study. First, limited by the detection performance of the detection module for small targets, our relative localization can currently only be achieved at a distance of less than 12 m. Of course, with the improvement in the detection performance, the action distance will be larger. Second, our currently generated position data has not been filtered. So based on the experimental conclusions, our next research direction is to improve the detection performance of the detection module for the motors as small targets at long distances, and the second is to improve the overall stability of the estimation value under the time series through the filtering algorithm.

**Author Contributions:** Conceptualization, X.S., F.Q. and M.K.; methodology, X.S. and M.K.; software, X.S. and G.X.; validation, M.K. and H.Z.; formal analysis, F.Q.; investigation, K.T.; resources, K.T.; data curation, G.X.; writing—original draft preparation, X.S.; writing—review and editing, F.Q. and M.K.; visualization, X.S.; supervision, F.Q.; project administration, M.K.; funding acquisition, H.Z. All authors have read and agreed to the published version of the manuscript.

**Funding:** This work was supported by The Natural Science Foundation for Young Scholars of Anhui Province under Grant No. 2108085QF255, The Research Project of National University of Defense and Technology under Grant No. ZK21-45, The Military Postgraduate Funding Project under Grant No. JY2022A006, and in part by The 69th Project Funded by China Postdoctoral Science Foundation under Grant No. 2021M693977.

**Data Availability Statement:** The data are available from the corresponding author on reasonable request.

**Conflicts of Interest:** The authors declare no conflict of interest.

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
