# Peer review of "Relative Localization within a Quadcopter Unmanned Aerial Vehicle Swarm Based on Airborne Monocular Vision"

_drones, doi:10.3390/drones7100612_

Round 1

Reviewer 1 Report

An airborne monocular vision-based relative localization technique based on a small quadrotor UAV is presented in this paper. The performances of proposed technique are evaluated in term of execution speed and localization accuracy using both simulation and real dataset.  The paper is written well. Sufficient experimental results are provided here to show that the proposed method achieves reliable accuracy in both simulation and real environment. However few issues are found in description of the proposed technique as stated below. Authors are suggested to address those issues.

1.     Appropriate reference for ‘YOLOv8-pose keypoint detection algorithm’ mentioned in Section 1 should be provided.

2.     A separate table in each of the Sections 3 & 4 must be provided to define various symbols & mathematical notations used in those two sections.  

3.     In subsection 4.1 (proposed model), three different coordinate systems, which are camera coordinate system, pixel coordinate system and motor coordinate system, are shown. It is to show how the above-said three coordinate systems are related to the world reference frame or Earth coordinate system.

4.     The ‘pinhole model imaging principle’ mentioned in subsection 4.2, either should be stated or appropriate reference should be provided.

5.       In subsection 4.3, two different coordinate systems for UAV and motors are shown. Since motor is a part UAV, it is required to explain why two different coordinates for them are considered here.

6.     The term ‘density’ shown in many figures in subsection 5.3 should be clearly defined.

Reviewer 2 Report

To immediately realize real-time relative localization between UAVs, the authors of this paper have developed a relative localization solution based on airborne monocular sensing data. To start, they used the simple YOLOv8-pose target identification method to recognize a quadcopter UAV and its rotor motors in real time. Then, in order to increase computing efficiency, they have fully utilized the UAV's geometrical features to create a more flexible method for addressing the P3P problem. When fewer than four motors are found, they have analytically suggested a positive solution determination strategy based on reasonable attitude information to tackle the multi-solution problem. In order to increase accuracy even more, they have added the majority of weights for the confidence in motor detection to the computation of relative localization location. Finally, they tested a prototype UAV using both simulation and actual experiments. They have demonstrated that the experimental data supports the viability of the suggested plan, in which the core algorithm performs noticeably better than the traditional approach. As a result, the work is worthy of publication if the remarks listed are taken into account:

1-             There are numerous researches that address the localization issue with UAVs and their accompanying outcomes. In these cases, different filtering techniques have been used. Examples include the widely utilized interacting multiple nonlinear fuzzy adaptive H models filter algorithms. So, is it conceivable to compare, say, IMM-NFAH or NH and the proposed one?

2-             The effectiveness and robustness of an autonomous vehicle's execution are significantly impacted by the control and navigation systems. Do you have any information about the robustness and execution performance of UAVs, as well as perhaps some comparisons? If this is the case or can be obtained, kindly include it in the paper's updated version.

3-             Even if the authors have investigated the previous studies I think it is still not clear the gap between the previous studies and the current one. Hence, the authors should clarify the gap between the existing research work and the work you intend to do.

4-             The authors have to clearly state the limitations of their study.

5-             Instead of using visual odometry or simultaneous localization for the Global Positioning System (GPS)-denied situations, mapping using 2D/3D cameras or laser rangefinders may be a better option. Can this case be conducted and a comparison made?

6-             There has been a lot of application of the machine learning (ML) strategy based on real-time measurements of radio frequency (RF) data in a practical environment. Different RF properties have been used to investigate various ML approaches, including decision trees (DT), support vector machines (SVM), and k-nearest neighbor (k-NN). The results demonstrate the superiority of a weighted k-NN, channel transfer function (CTF), and frequency coherence function (FCF)-based ML methodology over competing methods. Therefore, could you contrast the suggested method with the ones stated above and, if you can, offer a conclusion?

7-             The following articles can be considered and added to the introduction part of the study to improve the quality of the study. 1) 1D-LRF Aided Visual Inertial Odemetry for High-Altitude MAV Flight; 2) BOLD Bio-Inspired Optimized Leader Election for Multiple Drones; 3) Analysis of Wavelet Controller for Robustness in Electronic Differential of Electric Vehicles An Investigation and Numerical Developments; 4) Electric Power Components and Systems; 5) Design optimization of a fixed wing aircraft; 6) Intelligent Autonomous Systems 13; 7) Stereo Visual Inertial Mapping Algorithm for Autonomous Mobile Robot; 8) Computer Vision – ECCV.

8-             The conclusion should be written in bullet points with only the most important findings.

While the work is well-written in general, it, unfortunately, contains some grammatical and typographical problems. Before resubmitting the manuscript, it is suggested that the authors reread it.

Reviewer 3 Report

This paper presents airborne monocular vision-based relative localization 92 scheme using a small quadrotor UAV. The authors have clearly identified their contribution. Their novelty is quite sound and that worth for publication.

Author Response

Dear Reviewer:

Thank you very much for recognizing our article. We have improved the article based on the comments of the other two reviewers.

Sincerely yours,

Xiaokun Si

Round 2

Reviewer 2 Report

No further comments